# Morphological symmetries in robot learning

Daniel Ordonez-Apraez*, Mario Martin†‡, Antonio Agudo* and Francesc Moreno-Noguer*

* Institut de Robòtica i Informàtica Industrial, CSIC-UPC. †Barcelona Supercomputing Center (BSC).
‡ Departament de Ciències de la Computació, Universitat Politècnica de Catalunya (UPC).

[dordonez, aagudo, fmoreno]@iri.upc.edu, mmartin@cs.upc.edu

*Abstract*—This work studies the impact of morphological symmetries in learning applications in robotics. Morphological symmetries are a predominant feature in both biological and robotic systems, arising from the presence of planes/axis of symmetry in the system's morphology. This results in harmonious duplication and distribution of body parts (e.g., humans' sagittal/left-right symmetry). Morphological symmetries become a significant learning prior as they extend to symmetries in the system's dynamics, optimal control policies, and in all proprioceptive and exteroceptive measurements, related to the system's dynamics evolution [10]. Exploiting these symmetries in learning applications offers several advantageous outcomes, such as the use of data augmentation to mitigate the cost and challenges of data collection, or the use of equivariant/invariant function approximation models (e.g., neural networks) to improve sample efficiency and generalization, while reducing the number of trainable parameters. Lastly, we provide a video presentation[1] and an open access repository[2] reproducing our experiments and allowing for rapid prototyping in robot learning applications exploiting morphological symmetries.

## I. Introduction

Discrete Morphological Symmetries (DMSs) are ubiquitous in both biological and robotic systems. The vast majority of living and extinct animal species exhibit bilateral/sagittal reflection symmetry, where the right side of the body is approximately a reflection of the left side (see fig. 1-left). Similarly, a significant number of species exhibit radial symmetry, characterized by two or more morphological symmetry planes/axis (see fig. 1-center) [6]. These symmetries are a consequence of nature's tendency to symmetric body parts and harmonic duplication and distribution of limbs. A pattern perfected and exploited in the design of robotic systems.

Symmetries of the state of a dynamical system translate to symmetries of the system's dynamics and control [17]. Thus, DMSs imply the presence of symmetries in the dynamics and control of body motions, extending to symmetries in all proprioceptive and exteroceptive measurements, related to the evolution of the system's dynamics (e.g., joint position/velocity/torque, depth images, contact forces). Therefore, for systems with morphological symmetries, we can use data augmentation to mitigate the challenges of data collection in robotics, computer graphics, and computational biology. This, roughly implies that for every minute of recorded data of a system with $n$ morphological symmetries, we can obtain an additional $n - 1$ minutes of recordings, solely by considering the symmetric states of the recorded data. See the case of

the robot Solo in fig. 1-center, for which we obtain 3 additional minutes of recording by considering the depicted 4-fold symmetries. Furthermore, we can exploit the symmetries of proprioceptive and exteroceptive data by imposing symmetry constraints in machine learning algorithms to boost sample efficiency and enhance generalization [17, 4, 12]. Consider the case of robot Solo in fig. 1-center/right. We desire to approximate the function $y = f(x)$, mapping points in an input space $x \in \mathcal{X}$ (say, the state of our robot) to points in an output space $y \in \mathcal{Y}$ (say, the binary contact state of the robot's feet). To achieve this we use recorded data to train a function approximation model $\hat{f}$ parameterized with $\phi$, i.e. $y \approx \hat{f}(x; \phi)$. Because of the robot morphological symmetry, the input and output spaces have symmetries, and our target function is subjected to an equivariance constraint:

$$g \cdot y = f(g \cdot x) \mid \quad \forall \quad g \in \mathcal{G}. \tag{1}$$

Where $g$ represents a symmetry, $g \cdot x$ and $g \cdot y$ the input and output points transformed by the symmetry (in our example, $g \cdot x$ is the transformed robot state and $g \cdot y$ a different contact state), while $\mathcal{G}$ represents the set of symmetries of the robot, its symmetry group. In these scenarios, we should impose the same equivariance constraints of our target function (eq. (1)) to our model $\bar{f}$. Since by doing so, we are reducing the solution space of the optimization algorithm used to find the optimal $\bar{f}$. In practice, imposing equivariance (or invariance) constraints implies reducing the number of parameters of your model $\phi$, while empirically obtaining benefits in sample efficiency and generalization [4, 12, 10].

Despite the potential benefits of exploiting symmetry and the ubiquitous presence of morphological symmetries in robotic/biological/virtual systems, this relevant inductive bias is frequently left unexploited in data-driven applications in robotics, computational biology, and computer graphics. We attribute the scarce adoption of these techniques to a missing theoretical framework that consolidates the concept of morphological symmetries, facilitating their study and identification. And, to a missing practical framework enabling the efficient and convenient exploitation of symmetries in real-world data-driven applications.

The identification of morphological symmetries and how these extend to symmetries of proprioceptive and exteroceptive data is currently a laborious and error-prone system-specific process, due to the lack of a clear theoretical framework. As a result, most recent works that exploit some morphological symmetry (e.g., [15, 1, 16] in computer graphics and [12, 9, 5, 3] in robotics/dynamical systems) have only been applied

---

[1] Video presentation: youtu.be/qu4jIViRU1A
[2] Code repository: github.com/Danfoa/MorphoSymm

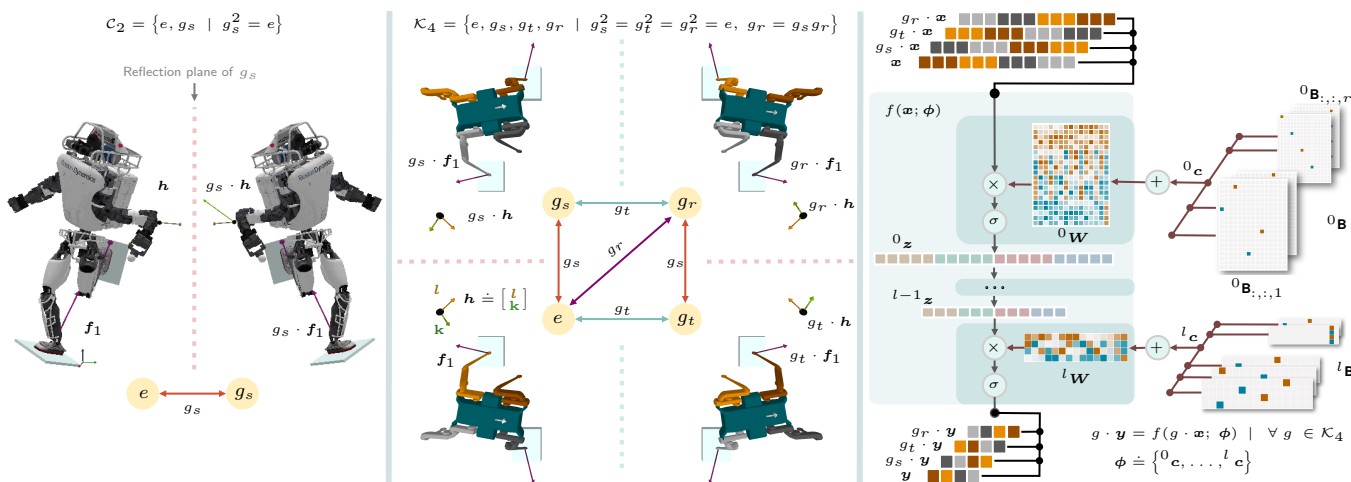

Fig. 1: **Left:** Symmetric configurations of the bipedal robot Atlas (3D animation) illustrating its morphological symmetry described by the reflection group $\mathcal{C}_2$. The robot can imitate the reflections $g_s$ (hint: note the non-reflected text on the robot's chest). **Middle:** Top-view of symmetric configurations of the quadruped robot Solo (3D animation) showcasing its morphological symmetries described by the Klein four-group $\mathcal{K}_4$. The robot can imitate two reflections ($g_s, g_t$) and a $180°$ rotation ($g_r$) of space (hint: observe the unreflected/unrotated robot's heading direction and legs coloring). Symmetry transformations (arrows) affect the robot's configuration, as well as proprioceptive measurements (center of mass linear $\boldsymbol{l}$ and angular $\mathbf{k}$ momentum) and exteroceptive measurements (terrain elevation, external force $\boldsymbol{f_1}$). **Right:** Diagram of a toy $\mathcal{K}_4$-equivariant neural network, processing the symmetric states of robot Solo $\boldsymbol{x}$ and outputting the symmetric binary foot contact states $\boldsymbol{y}$ (see section IV).

to simple systems and the simplest morphological symmetry: reflection/sagittal symmetry (see fig. 1-left), with the exception of Finzi et al. [3]. However, these works provide little guidance on how to apply these techniques to other systems, particularly those with more than a single morphological symmetry.

Our recent work [10], aims at increasing the adoption of morphological symmetry exploitation in robotics by presenting the theoretical and practical contributions[2] that enable the study and exploitation of these symmetries in arbitrary dynamical systems with any number of symmetries. In this short paper, we summarize the most important facts of morphological symmetries in robotics and their implications in data-driven applications. For a rigorous and extended development, we refer the interested reader to [10].

## II. PROPERTIES OF SYMMETRIC DYNAMICAL SYSTEMS

In robotics a symmetry $g$ is roughly defined as an energy-preserving transformation of the robot state $(\boldsymbol{q}, \dot{\boldsymbol{q}})$, defined by the system generalized position $\boldsymbol{q} \in \mathrm{Q}$ and velocity coordinates $\dot{\boldsymbol{q}} \in \mathrm{T}_{\boldsymbol{q}}\mathrm{Q}$. If a dynamical system has a group of symmetries $\mathcal{G}$, its dynamics (i.e, its equations of motion $\mathbf{M}(\boldsymbol{q})\ddot{\boldsymbol{q}} = \boldsymbol{\tau}(\boldsymbol{q}, \dot{\boldsymbol{q}})$) are equivariant. That is:

$$g \cdot [\underbrace{\mathbf{M}(\boldsymbol{q})\ddot{\boldsymbol{q}}}_{Inertial} - \underbrace{\boldsymbol{\tau}(\boldsymbol{q}, \dot{\boldsymbol{q}})}_{Moving}] = \underbrace{\mathbf{M}(g \cdot \boldsymbol{q})g \cdot \ddot{\boldsymbol{q}}}_{Inertial} - \underbrace{\boldsymbol{\tau}(g \cdot \boldsymbol{q}, g \cdot \dot{\boldsymbol{q}})}_{Moving} = \mathbf{0}$$
$$\mid \forall\, g \in \mathcal{G},\ \boldsymbol{q} \in \mathrm{Q},\ \dot{\boldsymbol{q}} \in \mathrm{T}_{\boldsymbol{q}}\mathrm{Q}. \quad (2)$$

Denoting $\mathbf{M}(\boldsymbol{q}) : \mathrm{Q} \to \mathbb{R}^{n \times n}$ as the generalized mass matrix function and $\boldsymbol{\tau}(\boldsymbol{q}, \dot{\boldsymbol{q}}) : \mathrm{Q} \times \mathrm{T}_{\boldsymbol{q}}\mathrm{Q} \to \mathbb{R}^n$ as the generalized moving forces at a given state $(\boldsymbol{q}, \dot{\boldsymbol{q}})$.

This property of symmetric dynamical systems, denoted as dynamics $\mathcal{G}$-equivariance (eq. (2)), depends on both the

generalized inertial and moving forces being independently equivariant, implying:

$$\mathbf{M}(g \cdot \boldsymbol{q}) = g\mathbf{M}(\boldsymbol{q})g^{-1} \quad \wedge \quad g \cdot \boldsymbol{\tau}(\boldsymbol{q}, \dot{\boldsymbol{q}}) = \boldsymbol{\tau}(g \cdot \boldsymbol{q}, g \cdot \dot{\boldsymbol{q}})$$
$$\mid \forall\, g \in \mathcal{G},\ \boldsymbol{q} \in \mathrm{Q},\ \dot{\boldsymbol{q}} \in \mathrm{T}_{\boldsymbol{q}}\mathrm{Q}. \quad (3)$$

The equivariance of the inertial forces requires that the generalized mass matrix of the systems is equivariant. This is the identifying property of symmetrical dynamical systems. In practice, as the generalized mass matrix is well-defined for model-based systems, it can be used for the identification of system's symmetries using eq. (3) (see [10] for the case of rigid body dynamics). Furthermore, the equivariance of the generalized moving forces (which in practice, usually incorporates control, constraint, and external forces) implies that dynamics $\mathcal{G}$-equivariance (eq. (2)) is upheld until a symmetry breaking force violates the equivariance of $\boldsymbol{\tau}$.

To gain some intuition, consider as an example the bipedal robot Atlas, with symmetry group $\mathcal{G} = \mathcal{C}_2 = \{e, g_s\}$. Both robot states in fig. 1-left are symmetric states (related by the action $g_s$). Then, eq. (2) suggests that any trajectory of motion, starting from the left robot state, will be equivalent (up to transformation by $g_s$) to a motion trajectory starting from the right robot state, if and only if, the moving forces driving both trajectories are equivalent (up to transformation by $g_s$). That is if the control and external forces are $\mathcal{C}_2$-equivariant (eq. (3)). Note, we can perform a similar analysis for each symmetric state and action of systems with larger symmetry groups (e.g. Solo in fig. 1-center).

The aforementioned definition of symmetries as energy-preserving transformations of the system state is intentionally generic, imposing no restrictions on the nature of the state

transformation, such as whether the transformed state is feasible or reachable. This allows us to consider feasible state transformations (such as robot translations and rotations[3]) along with unfeasible state transformations (such as a reflection of space) as symmetries of the system. Naturally, in robotics, we are interested in studying and exploiting feasible symmetries alone. Therefore we introduced the concept of discrete morphological symmetry, as the set of feasible symmetries of the system that imitate feasible and unfeasible symmetries.

## III. DISCRETE MORPHOLOGICAL SYMMETRIES (DMSs)

A dynamical system is said to possess a DMS if it can imitate the effects of a rotation, reflection, or translation in space (i.e. Euclidean isometries), through a feasible discrete change in its configuration (see formal definition in [10]). To gain intuition, we can analyze the simplest and most common DMS.

*Reflection DMS:* Although most floating-base dynamical systems are symmetric with respect to reflections of space (section II), these symmetries are infeasible due to the impossibility to execute reflections in the real-world [11]. However, systems with sagittal symmetry (e.g., Atlas in fig. 1-left, or humans) can imitate the effect of a reflection with a feasible discrete change in their configuration, by rotating their body and modifying their limbs' pose. These systems share the same symmetry group, the reflection group $\mathcal{G} \equiv \mathcal{C}_2$.

*Multiple DMSs:* This property can be extended to the case of a floating-base system having multiple DMSs, allowing it to imitate multiple distinct Euclidean isometries. Most frequently systems can imitate a set of rotations and reflections, making $\mathcal{G}$ a Cyclic $\mathcal{C}_k$ or Dihedral $\mathcal{D}_{2k}$ group. See examples for $\mathcal{C}_3$ in [10], and for $\mathcal{D}_4 \equiv \mathcal{K}_4$ in fig. 1-center.

Because each DMS is defined as a feasible transformation that imitates a system's symmetry $\bar{g}$ due to a Euclidean isometry, the group of DMSs $\mathcal{G}$ is isomorphic to a subset of the feasible and unfeasible symmetries of the dynamical system due to rotations, reflections, and translations in space. Furthermore, the existence of the DMSs is subjected to the system's generalized mass matrix being $\mathcal{G}$-equivariant (eq. (3)). In practice, these constraints translate to identifiable constraints in the kinematic and dynamic parameters of the system model [10].

## IV. $\mathcal{G}$-EQUIVARIANT AND $\mathcal{G}$-INVARIANT FUNCTION APPROXIMATORS

Once we identified the DMS group $\mathcal{G}$ of our system, we know that any proprioceptive or exteroceptive measurements have the same symmetry group $\mathcal{G}$. Therefore, to improve generalization and sample efficiency, we can exploit the known symmetries of the input $\boldsymbol{x}$ and output $\boldsymbol{y}$ spaces, of any mapping we desire to approximate, by constructing $\mathcal{G}$-equivariant or $\mathcal{G}$-invariant (eq. (1)) function approximation models $\hat{f}(\boldsymbol{x}; \boldsymbol{\phi})$, parameterized with $\boldsymbol{\phi}$. In [10] we study

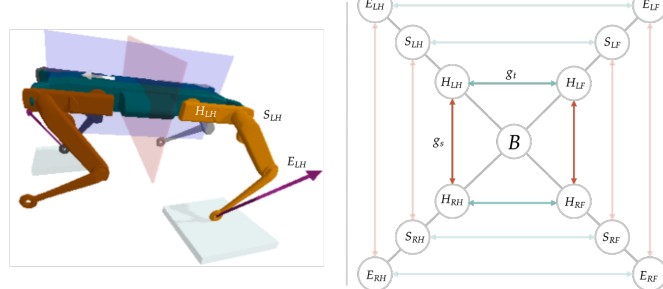

Fig. 2: **Left:** Solo sagittal (blue) and transversal (red) symmetry planes of the base body. **Right:** Solo's kinematic tree, and permutation symmetries of the legs/tree-branches.

the case of $\mathcal{G}$-equivariant/invariant neural networks (NN). In this section, we summarized the most relevant implications of DMSs for this type of machine-learning model.

- **Computational implications of using $\mathcal{G}$-equivariant NN.** Thanks to recent theoretical and practical developments [4, 10, 12], the use of $\mathcal{G}$-equivariant NN instead of unconstrained NN comes at the price of a negligible increase in memory and computational resources required during training of the model. Most importantly, there is no difference, at inference time, between equivariant and unconstrained models.
- **Number of trainable parameters of a NN.** Imposing equivariance/invariance constraints in NN signifies the reduction in the number of trainable parameters of the model [4, 12, 2]. In practice, this implies that for a $\mathcal{G}$-equivariant layer the number of trainable parameters is reduced by approximately $^1/|\mathcal{G}|$ being $|\mathcal{G}|$ the number of symmetries of the data (i.e., number of DMSs of the system). Therefore a $\mathcal{G}$-equivariant architecture with $\mathcal{G} = \mathcal{C}_2$ (robot Atlas in fig. 1-left), or $\mathcal{G} = \mathcal{K}_4$ (Solo in fig. 1-center) will have approximately $^1/_2$ (Atlas) or $^1/_4$ (Solo) of the trainable parameters of an unconstrained NN of the same architectural size. The reduction of parameters is caused by the parameter sharing and is visually depicted in fig. 1-right.

An increasing amount of theoretical [2, 14] and empirical [12, 3, 10, 13] evidence suggest that when the data features symmetries, the use of equivariant/invariant function approximation models leads to increase generalization capabilities and a reduction in sample complexity. On [10] we present empirical evidence in robotics in a synthetic and real-world learning application. Here, we summarize the results of the real-world application.

## V. EXPERIMENTS

We present a supervised experiment using real-world data in a classification application to showcase the effectiveness of Discrete Morphological Symmetries (DMSs) for data augmentation and training equivariant functions. The goal is to demonstrate the positive impact of exploiting DMSs on the model's sample efficiency and generalization capacity. For a detailed analysis of the technical aspects and additional experiments, please refer to [10].

---

[3]In conservative systems, translational, rotational, and time-shift symmetries imply, by Noether's theorem, the conservation of linear momentum, angular momentum, and energy, respectively [8].

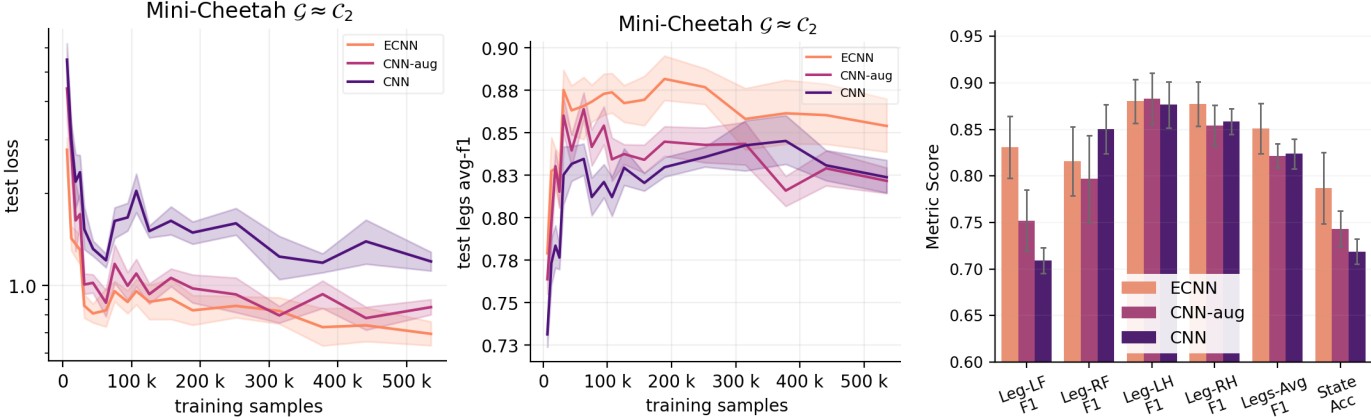

Fig. 3: **Static-Friction-Regime contact detection results comparing CNN, CNN-aug, and ECNN. Left:** Sample efficiency in log-log scale. **Middle:** Average legs F1-score. **Right:** Classification metrics on test set performance of models trained with the entire training set. The selected metrics include contact-state ($\boldsymbol{y} \in \mathbb{R}^{16}$) accuracy (Acc) and f1-score (F1) for each leg binary contact state. Due to the sagittal symmetry of the robot, the left front (LF) and right front (RF) legs are expected to be symmetric, as well as the left hind (LH) and right hind (RH) legs. F1-score is presented considering the dataset class imbalance (see [10]). The reported values represent the average and standard deviation across 8 different seeds.

### A. Static-friction-regime contact detection (Classification)

In this experiment, we utilize the dataset introduced in Lin et al. [7] for estimating static-friction-regime contacts in the foots of the Mini-Cheetah quadruped robot. The dataset consists of real-world proprioceptive data ($\hat{\boldsymbol{q}}$, $\dot{\hat{\boldsymbol{q}}}$, base linear acceleration, base angular velocity, and leg feet positions and velocities) captured over a history of $150$ time-frames. These measurements were obtained from inboard sensors during locomotion, encompassing various gaits and terrains. The dataset also includes $\boldsymbol{y} \in \mathbb{R}^{16}$, representing the ground truth contact state of the robot, which was estimated offline using a non-causal algorithm. Our goal is to train a causal function approximator $\hat{f}(\boldsymbol{x}; \boldsymbol{\phi})$ to predict the contact state based on the input proprioceptive data.

The Mini-Cheetah robot in the real-world exhibits an approximate reflection symmetry group, $\mathcal{G} \approx \mathcal{C}_2$. As a result, both the proprioceptive data $\boldsymbol{x}$ and the contact state $\boldsymbol{y}$ share the symmetry group $\mathcal{G}$. In this experiment, we compare three variants of function approximators: the original Convolutional Neural Network architecture proposed by Lin et al. [7] (CNN), a version of CNN trained with data augmentation (CNN-aug), and a version of CNN that incorporates hard-equivariance constraints (E-CNN).

The sampling efficiency and average leg contact state classification results are depicted in fig. 3-left-&-middle. The equivariant model, E-CNN, demonstrates superior generalization performance and robustness to dataset biases compared to the unconstrained models [10]. Following E-CNN, CNN-aug exhibits better performance than the original CNN. In fig. 3-right, we evaluate the classification metrics of the test set when using the entire training data. The E-CNN model outperforms both CNN-aug and CNN in contact state classification and average leg contact detection. Notably, exploiting symmetries helps mitigate suboptimal asymmetries in the models, preventing them from favoring the classification of one leg over others

(observe legs LF and RF in fig. 3-right).

## VI. Conclusions & Discussion

In this work, we summarize the findings presented in [10], where we present the definition of Discrete Morphological Symmetry (DMS): a capability of some dynamical systems to imitate the effect of rotations, translations, and infeasible reflections of space with a feasible discrete change in the system configuration. Using the language of group theory we study the set of DMSs of a dynamical system as a symmetry group $\mathcal{G}$ and conclude that: (1) A system with a symmetry group $\mathcal{G}$ exhibits $\mathcal{G}$-equivariant generalized mass matrix and dynamics. (2) That the symmetries of the dynamics extend to optimal control policies as well as to any proprioceptive and exteroceptive measurements, related to the evolution of the system's dynamics.

We establish the necessary theoretical abstractions to investigate and identify DMSs in any dynamical system, irrespective of the number of symmetries present. This new formalism allows us to identify the reflection/sagittal symmetry, prevalent in humans, animals, and most robots, as the simplest morphological symmetry group $\mathcal{G} = \mathcal{C}_2$. Crucially, we use the same formalism to identify and exploit DMSs in real-world robotic systems with a greater number of symmetries.

In addition, we provide an open-access repository that facilitates the efficient prototyping of $\mathcal{G}$-equivariant neural networks for exploiting DMS in various applications involving rigid-body dynamics, such as robotics, computer graphics, and computational biology. This repository includes a growing collection of symmetric dynamical systems, with their corresponding symmetry groups already identified. Furthermore, we present compelling empirical and theoretical evidence supporting the utilization of DMSs in data-driven applications through data augmentation and the adoption of $\mathcal{G}$-equivariant neural networks. Both symmetry exploitation techniques result in improved sample efficiency and generalization.

ACKNOWLEDGMENTS

This work's experiments were run at the Barcelona Super-computing Center in collaboration with the HPAI group. This work is supported by the Spanish government with the project MoHuCo PID2020-120049RB-I00 and the ERA-Net Chistera project IPALM PCI2019-103386.

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
