# OpenReview forum: "Morphological symmetries in robot learning"
_roboticsfoundation.org/RSS/2023/Workshop/Symmetry — RSS 2023 Workshop Symmetry_

### Official Review · Reviewer_7c6C · 2023-06-10
**Accept**

**Rating:** 7
**Confidence:** 4

**Review:**

This paper propose to take advatnage of Discrete Morphological Symmetries (DMSs) which widely exist in robotic systems in the context of robot learnining. More specifically, the authors propose the explot DMSs via data augmentation or using neural network architectures with built-in structure for equivariance/invarance informed by DMSs. The authors enumerate common DMSs in robotic systems using the language of Group Theory and establishes a new formalism to identify DMSs. Using experiment results on the foot contact classification task for quadruped robots, the authors demostrate the advatage of explorting DMSs in terms of sample efficiency for both data agumetnation and equivariant network architecture designs.

The paper is clearly written with claims supported by experiences on real-world collected robotic datasets.

---

### Official Review · Reviewer_qiXF · 2023-06-16
**Weak Accept**

**Rating:** 6
**Confidence:** 4

**Review:**

This paper studies a predominant feature --morphological symmetries in robot system dynamics. Examples of robot morphological symmetries are given and investigated. To exploit the symmetries, the paper engineered data augmentation method and equivariant neural networks. Experiment of extroceptive states detection verified the advantage of data augmentation and equivariant neural network.

The cons are the paper only performed experiments on one robot morphology configuration and is limited to extroceptive state detection, rather than robotic control policy learning.

---

### Decision · Program_Chairs · 2023-06-23

**Decision:**

Accept

**Comment:**

Congratulations! We encourage the authors to revise the paper based on the reviewer's feedback.
Your paper will be presented as both a short presentation and a poster. Detailed instructions about the presentation format and camera-ready submission will be sent to you soon.